# Bio-Based Polyurethane Foams from Kraft Lignin with Improved Fire Resistance

**DOI:** 10.3390/polym15051074

**Published:** 2023-02-21

**Authors:** Fernanda R. Vieira, Nuno V. Gama, Dmitry V. Evtuguin, Carlos O. Amorim, Vitor S. Amaral, Paula C. O. R. Pinto, Ana Barros-Timmons

**Affiliations:** 1CICECO-Institute of Materials and Department of Chemistry, Campus de Santiago, University of Aveiro, 3810-193 Aveiro, Portugal; 2CICECO-Institute of Materials and Department of Physics, University of Aveiro, 3810-193 Aveiro, Portugal; 3RAIZ, Forest and Paper Research Institute, Quinta de S. Francisco, 3801-501 Aveiro, Portugal

**Keywords:** lignin, oxyalkylation, bio-based polyol, polyurethane foam, thermal insulation, thermal conductivity, fire reaction

## Abstract

Rigid polyurethane foams (RPUFs) were synthesized using exclusively lignin-based polyol (LBP) obtained via the oxyalkylation of kraft lignin with propylene carbonate (PC). Using the design of experiments methodology combined with statistical analysis, the formulations were optimized to obtain a bio-based RPUF with low thermal conductivity and low apparent density to be used as a lightweight insulating material. The thermo-mechanical properties of the ensuing foams were compared with those of a commercial RPUF and a RPUF (RPUF-conv) produced using a conventional polyol. The bio-based RPUF obtained using the optimized formulation exhibited low thermal conductivity (0.0289 W/m·K), low density (33.2 kg/m^3^), and reasonable cell morphology. Although the bio-based RPUF has slightly lower thermo-oxidative stability and mechanical properties than RPUF-conv, it is still suitable for thermal insulation applications. In addition, the fire resistance of this bio-based foam has been improved, with its average heat release rate (HRR) reduced by 18.5% and its burn time extended by 25% compared to RPUF-conv. Overall, this bio-based RPUF has shown potential to replace petroleum-based RPUF as an insulating material. This is the first report regarding the use of 100% unpurified LBP obtained via the oxyalkylation of LignoBoost kraft lignin in the production of RPUFs.

## 1. Introduction

Polymeric foams are widely used in various engineering applications due to their unique combination of properties, such as low density, good thermal insulation, high impact, and resistance. This type of material is typically prepared using a blowing agent and a polymeric matrix to yield a cellular structure with numerous small pockets of gas. Due to their textural characteristics, they are often used as thermal insulation materials in buildings, in refrigeration equipment, in automotive applications, cushioning and packaging, and in structural applications. Examples of polymeric matrices commonly used include thermoplastics such as poly(vinyl chloride) (PVC), polystyrene, and poly(acrylonitrile butadiene styrene) (ABS) as well as crosslinked polymers, with polyurethanes being a prime example of the most widely used of this kind [1]. Polyurethane foams (PUF) present several advantages, for example, excellent thermal insulation properties, good durability, and versatility in the formulation. Although there are several types of PUF such as flexible, rigid, and elastomeric, the most efficient high-performance insulation materials are RPUFs [2]. However, so far, the majority of commercially available RPUFs are derived from petroleum-resources. Therefore, extensive efforts have been made to develop raw materials that can at least partially replace the petroleum-based materials [2,3].

Among the several renewable raw materials that can be used to produce RPUFs, kraft lignin (KL) is most promising as it is a large-scale by-product of the pulp and paper industry, which is present in the black liquor and consists in alkali-soluble aromatic oligomers. According to common practice, kraft lignin is burned to supply heat and power for the pulping process, yet, it can be used as a renewable source of polyols [4,5]. KL can be isolated from black liquor using the commercial LignoBoost^®^ process to afford a high material purity with reasonably high molecular weight, rich in phenolic and aliphatic hydroxyl groups (OHs), high carbon content, and acceptable thermal stability [6,7]. These inherent characteristics of kraft lignin make it a potential polyol to be used in the production of RPUFs.

Even though the OH groups of isolated lignin can react with the isocyanates to yield polyurethanes (PU), its direct use remains a challenge due to their low reactivity (especially phenolic OHs), aggregation, and poor solubility of isolated lignin in other polymeric matrices [8,9]. This can be overcome by liquefaction processes which improves the accessibility of lignins’ OH groups and converts its phenolic OHs into more reactive aliphatic OH groups. Among established liquefaction processes, oxyalkylation using cyclic carbonates requires mild conditions when compared to the conventional liquefaction method (e.g., oxyalkylation with propylene oxide) [10,11,12,13,14] and the ensuing oxyalkylated lignin has a strong potential to be used as a liquid polyol. However, the influence of the oxyalkylation process on the main properties of the polyol, namely hydroxyl number (I_OH_) and viscosity, which have a major impact on PU formulations, requires systematic studies for each particular polyol. To date, information on the effect of oxyalkylated lignin polyols and subsequent formulations on the thermal insulation performance of RPUFs is still scarce. For example, Zhang et al. [13] studied the effect of oxyalkylation conditions of lignin using ethylene carbonate and polyethylene glycol (PEG) 400 on the density and compressive strength of RPUFs but the thermal conductivity of foams was not evaluated. In addition, the conditions used in this process were not so mild. An important contribution to this problem was the work of Duval et al. [15] who have investigated the effect of different contents of crude lignin-based polyol (LBP) obtained via the oxyalkylation of organosolv lignin using ethylene carbonate and PEG on the quality of RPUFs. As the crude LBP showed rather high reactivity the preparation of large samples of foam were compromised due to the very fast foam expansion, even upon reducing the amount of catalyst used. For that reason, the authors decided to limit the substitution of conventional petroleum derived polyol by only 25% of LBP. The ensuing foams presented similar properties to commercial RPUFs, namely thermal conductivity values (λ) of 0.0250 W/m·K. Recently, the oxyalkylation conditions of Lignoboost kraft lignin using propylene carbonate (PC) have been optimized to produce crude LBP with suitable I_OH_ and viscosity for the production of different PU products such as rigid foams and adhesives [16] and to develop formulations to prepare RPUFs for thermal insulation [17]. Using the design of experiments (DoE) methodology and subsequent statistical analysis of the data obtained, it was concluded that the blowing agent (BA) played a crucial role, and that the amount of surfactant needed to be included as a process variable in order to achieve better results.

The importance of the characteristics of the polyols, and of the formulation in general, in producing RPUFs for thermal insulation, is due to the fact that RPUFs consist of a solid network formed by the reaction between the hydroxyl groups of polyol and isocyanate groups of polyisocyanates, and a gaseous phase consisting of blowing agents [18,19]. Only when a fine balance between all these reagents is found, is it possible to obtain RPUFs with low thermal conductivity and density, good dimensional stability, and adequate strength [2,20]. Specifically, for thermal insulation applications, the most important property is the thermal conductivity, which should be below 0.030 W/m·K [21,22]. Hence, besides the characteristics of the main components, i.e., polyol and polyisocyanate, among the reagents used in the formulation of foams, the BA is of paramount importance, since the entrapped gas in the RPUF corresponds to 95–97% of its volume and is responsible for more than 50% of the thermal conductivity [1]. Whilst water is a commonly used BA in the production of CO_2_ (λ = 0.0145 W/m·K at 20 °C), for insulating applications the use of hydrocarbons (HCs), such as *n*-pentane, has emerged as a suitable alternative, since its global warming potential (GWP) is nearly zero and its use in foams has shown excellent insulating properties [23]. This is not only due to the lower thermal conductivity of *n*-pentane (λ = 0.0137 W/m·K at 20 °C), but especially because of the highly exothermic nature of the reaction leading to CO_2_ formation which over-accelerates the foam expansion process resulting in cellular disruption. Nonetheless, the use of HCs has its drawbacks such as flammability and low solubility in polyols or other PU raw materials [24]. To circumvent the solubility issue, the use of appropriate surfactants can be a solution [25]. Yet, the choice of surfactant needs to be very judicious as surfactants are responsible for stabilizing and controlling the size and the dispersity of the gas bubbles in the initial reactive foaming mixture as well as the final cell size of RPUFs. Indeed, the shape and size of cell are important in the thermal conductivity of the RPUF since the thermal conductivity can be lowered by reducing their cell size [26]. In the production of RPUFs, a commonly used type of surfactant are copolymers of polysiloxane and polyether whose ratio, number of polyether pedant chains attached to the silicone backbone, and the type of capping group determine the properties and, subsequently, their effect on the RPUF’s characteristics [25]. Considering that the formation of foams involves gelling and blowing (gas) reactions which have a direct impact on the polymerization reaction and expansion of the crosslinked structure, the type and amount of catalysts used also needs to be carefully determined in order to have control over the foams’ structure [20]. 

In an effort to contribute to the sustainable development the main purpose of this study was to optimize the formulation of bio-based RPUF, derived from lignin, to produce lightweight foams with low thermal conductivity for thermal insulation, namely in buildings. The effect of different variables such as the blowing agent, catalyst, and surfactant content on the thermal conductivity and density of foams was investigated using a DoE. The properties of the ensuing bio-based RPUF were compared with those of a commercial RPUF and of a RPUF prepared using a conventional polyether polyol (RPUF conv). Furthermore, the fire behavior of the bio-based RPUF was compared with that of (RPUF conv) to assess if the structural characteristics of lignin (known to enhance fire retardancy) could improve the fire resistance of the ensuing foam.

## 2. Materials and Methods

### 2.1. Materials

For the synthesis of LBP, LignoBoost kraft lignin (KL), propylene carbonate (PC), and a catalyst 1,8-diazabicyclo [5.4.0] undec-7-ene (DBU) were used as major ingredients. The KL obtained by LignoBoost procedure from eucalyptus black liquor was kindly supplied by a Portuguese pulp mill. PC was purchased from Acros Organics Comp. and was used without any further purification. DBU was supplied by Aldrich Chemical Comp. and used as received.

For the synthesis of RPUFs, Alcupol R^®^-2510 with hydroxyl number (I_OH_) of 250 mg KOH/g and viscosity at 25 °C of 0.25 Pa. s was supplied from Repsol and used as polyether-polyol to produce the conventional polyol-based foam (RPUF-conv). The LBP presents I_OH_ of 257 mg KOH/g, viscosity of 5.3 Pa. s, water content of 0.08%, containing circa 15.5% of PC-oligomers was obtained by the oxyalkylation process and was used without purification following the procedures described elsewhere [12,16]. The polymeric isocyanate Voranate M229 MDI with an NCO content of 31.1% and viscosity of 0.19 Pa. s was supplied by Dow Chemicals. The catalyst used in this work was DMCHA supplied by Chemical Comp. Tegostab^®^ B84501, and the polyether-modified polysiloxane surfactant was kindly supplied by EVONIK Chem. Comp. A mixture of *n*-pentane and water was used as blowing agent. The *n*-pentane used in this work by Sigma-Aldrich. A commercial RPUF was kindly supplied by SOPREMA Chem. Comp. (Strasbourg, France).

### 2.2. Rigid Polyurethane Foam Formulation

RPUFs were prepared by a two-component system: different amounts of surfactant, catalyst, and blowing agent (a mixture of 2% of water and different quantities of *n*-pentane) were added to a paper cup containing the polyol. The resulting mixture (component A) was homogenized using an IKA Ost Basic mixer with rotating blades for ca. 20 s at 900 rpm. The pMDI (component B) was mixed with the component A for 20 s at 900 rpm, in such quantities to ensure NCO index = 1.2. It should be noted that the amounts of water present in the polyols were subtracted from the amounts of blowing agent added to give the appropriate NCO/OH ratio. The RPUFs were obtained under free rise conditions. Besides the foams prepared using LBP, a RPUF-conv was also produced for comparison purposes. The formulation used was based on the experience of our group and some preliminary tests which included visual inspection of the ensuing foams to ensure absence of holes and dimensional stability. Furthermore, the criteria of low thermal conductivity and low density were also considered. Likewise, the methodology used in the production of foams was based on our previous experience [17,27]. All formulations are listed in Table 1. 

### 2.3. Rigid Polyurethane Foam Characterization

The thermal conductivity was measured at room temperature using the Transient Plane Source method with a Hot Disk^®^ Thermal Constant Analyzer TPS 2500S [28]. The Kapton sensor C5501 (radius 6.4 mm) used a 40 s measuring time and 8 mW heating power, and was placed between the plane surfaces of two pieces of sample to have uniform heat dissipation from both sides of the sensor during the measurements. Five measurement repetitions were made for each sample.

The foams were cut in cubic form (1 × 1 × 1 cm^3^) and weighted to determine the apparent density by dividing the weight of the foams by the calculated volume. The values of density correspond to the average density determined for 10 specimens of each foam.

The surface morphology of foams was evaluated using a SU-70 (Hitachi) scanning electron microscope (SEM). The foams were coated with gold to avoid electrostatic charging during examination and analyzed using accelerating voltage of 15.0 kV. The SEM images were recorded along the direction of foam rise and the software Image J was used to obtain the average cell size. The average cell diameter was estimated from more than 100 measurements of each foam.

The closed cells content was determined using a gas pycnometer instrument according to ASTM D6226-05 standard [29]. The foams were cut in circular form and their volume was measured to be used in the calculation of the closed cell content. 

The Brunauer, Emmett, and Teller (BET) surface area and pore volume were determined using a surface area analyzer (Quantachrome- Autosorb IQ2) and nitrogen gas. The samples were outgassed overnight prior to adsorption measurements. The BET model was used to fit the adsorption isotherm data and to calculate the specific surface area of the samples. The pores diameters and their volume were calculated according to the model developed by Barret, Joyner, and Halenda (BJH) [30]. Three measurements were made for each sample.

The mechanical tests were performed using a Instron 5966 universal machine equipped with a 1 kN load cell according to the ASTM D1621 standard [31]. The foams were placed between the two parallel plates and compressed at 2.5 mm/min up to 30% deformation. The analysis was performed along the direction of foaming and the values presented correspond to the average of five specimens. 

The thermogravimetric analysis (TGA) was performed to evaluate the thermo-oxidative degradation of the foams using a SET-SYS Evolution 1750 thermogravimetric analyzer (Setaram), ISO 11357 standard [32], from room temperature up to 800 °C, at a heating rate of 10 °C/min under oxygen flux (200 mL/min). 

Dynamic mechanical analyses (DMA) were carried out using a Tritec 2000 equipment (Triton Technologies) in compression mode using multi frequency and temperature scan. Specimens with dimensions of 10 mm × 10 mm × 7 mm were analyzed from −55–250 °C at a constant heating rate of 2 °C.min^−1^, at 1 Hz and 10 Hz, applying a displacement amplitude of 0.020 mm. 

The fire reaction performance of the RPUFs was analyzed using a cone calorimeter test in an external laboratory, INEGI- University of Porto. RPUF specimens (100 × 100 × 50 mm) were tested using a heat flux of 50 kW/m^2^. Prior to testing, the specimens were conditioned for a period of 96 h at 23 ± 2 °C and 50 ± 5% relative humidity, to ensure the constant mass criterion. The conditioned specimens were wrapped in a single layer of aluminum foil, with the shiny side towards the specimen. The average heat release rate (HRR) and the MARHE (maximum averaged rate of heat emission) were determined according to the ISO 5660 standard. Three measurements were made for each sample.

## 3. Results and Discussion

### 3.1. Effect of Blowing Agent, Catalyst and Surfactant Content on the Density and Thermal Conductivity of Bio-Based Rigid Polyurethane Foam

Further to our previous studies on the development of formulations to prepare RPUF for thermal insulation using LBP derived from the oxyalkylation of KL, in this study a more extensive evaluation of the effect of the amount of BA, surfactant and catalyst on the density and thermal conductivity of LBP derived RPUFs was carried out [17]. In order to maximize the amount of bio based polyol in the formulations, 100% crude LBP was used. Moreover, being the potential application of the RPUF as lightweight insulating material the target thermal conductivity should be ≤0.030 W/m·K and apparent density values should fall within the 25–60 kg/m^3^ interval. 

Besides the required replacement of water as BA by mixtures of *n*-pentane and water identified in the previous work [17], the proportion between those components had to be tuned. Additionally, to ensure adequate solubilization of *n*-pentane in the polyol and to help to stabilize and control the size and the dispersity of the air bubbles in the initial reactive foaming mixture, as well as the final cell size of RPUFs, a polysiloxane-polyether copolymer was used as surfactant in quantities between 1 and 3% (relative to the mass of polyol). Moreover, as the catalyst used in previous studies was DABCO, which is considered a gelling catalyst [20], in the present study DMCHA was used because of its known ability to balance gelling and blowing reactions. Despite of the fact that our crude LBP contains a residual amount of the catalyst DBU (which is considered a gelling and blowing catalyst), the amounts of DMCHA considered in the present study varied between 0.5 and 1.5% (relative to the mass of polyol). Using the JMP^®^ software, a central composite design (CCD) was used to generate 16 formulations. The experimental ranges of the process variables (blowing agent, catalyst, and surfactant) and their characteristics were based on values available in the literature and the formulation experience of our group. The thermal conductivity and density values as well as the quality assessment of foams based on visual inspection are listed in Table 2.

The data were analyzed using the main effects and interaction effects of process variables on the responses in a hierarchical way using the Pareto chart to evaluate the relationship between the responses and the process variables. The Pareto chart depicted in Appendix A showed that only the linear effect BA, the quadratic effects of surfactant (SURF*SURF) and blowing agent (BA*BA), were statistically significant, as the *p*-value was smaller than 0.0500. Furthermore, the model fit, and adequacy was evaluated by the analysis of variance (ANOVA) (See Appendix A). Although the coefficients of determination (R^2^) were reasonably high, 0.80 and 0.81 for thermal conductivity and density, respectively, the quality of a statistical model depends on other statistical parameters such as, the F ratio and the *p*-value of models, and in our case the *p*-values were higher than 0.100. In view of these results, the models were reduced in a hierarchical way, eliminating insignificant terms. This procedure, of the elimination of insignificant terms, helps to improve and simplify the models. Table 3 presents the ANOVA results for the reduced models. The fact that the F ratio increased and the *p*-values were less than 0.0500 confirms the fitting and adequacy of the models [33]. In addition, even though the R^2^ values (0.730 for thermal conductivity and 0.722 for density) are lower than those obtained for the extended empirical models (See Table 3 and Appendix A), the residuals’ plot for the predicted thermal conductivity and density are uniformly distributed (See Appendix A). Yet, whilst the reduced models indicate that the independent variables are correlated with responses, they do not explain well the variability in the responses, especially in the case of the thermal conductivity. Indeed, for the thermal conductivity only the quadratic effects of surfactant (SURF*SURF) and blowing agent (BA*BA) and the interaction effects (BA*SURF) showed significant effects (See Appendix A). Hence, to try to understand how the predicted responses change as a function of the individual process variables, the prediction profiler, shown in Figure 1, was used. 

It is well known that both thermal conductivity and density are affected by the amount of BA [27,34,35], which contributes 50–70% of the overall foam thermal conductivity [36]. However, according to our results, no significant effect was registered. This might be attributed to the low solubility of *n*-pentane in the polymer matrix [23,37], and some evaporation of *n*-pentane during the exothermic gelling and blowing reactions, which may have compromised the nucleation of bubbles. Furthermore, according to Figure 1, the higher thermal conductivity values are associated with medium amounts of *n*-pentane. Again, this was not expected since it is well established in the literature that the thermal conductivity decreases as the amount of physical blowing agent increases in the formulation [34,38]. Nonetheless, it should be taken into consideration that other parameters also influence the thermal conductivity such as the average cell size, density, closed cell content, and the viscosity of polyols, which in fact are all interconnected. For example, the viscosity of LBP is 5.3 Pa. s, while that of the conventional polyol is 0.25 Pa.s, hence, the higher viscosity of LBP was expected to lead to the reduction in cell size and consequently reduce the thermal conductivity values, since it can limit bubble coalescence and expansion [20,39,40,41,42]. Conversely, high values of viscosity can cause coarser and less uniform cells. Another characteristic of crude LBP is the presence of PC-oligomer and residual catalyst. Hence, the very reactive crude LBP formulation may cause disturbances in the gelling and blowing reactions, causing the partial collapse of the foam, and consequently can reduce the closed cell content of the foam and consequently increase the thermal conductivity values.

With regard to the impact on density, it is observed that it decreases as the amount of blowing agent increases. The same trend has been reported in the literature [27,34,38,43], and is attributed to the fact that higher amounts of blowing agent increase the size of bubbles, thus decreasing the density of foams.

Parallel to the statistical analyses, based on the visual inspection, three foams were chosen to evaluate their morphology and textural properties, in order to get a better insight into the influence of formulations on the foams’ quality. Table 4 summarizes the characteristics of these foams. Analyzing the cellular structure presented in Figure 2, foam RPUF-1 presents a more homogeneous cellular structure. In turn, foams RPUF-6 and RPUF-11 have lower thermal conductivity and density and slightly larger average cell size than RPUF-1. This can be attributed to bubble coalescence due to the high amounts of blowing agent used [44,45]; despite the fact that a higher amount of surfactant was used in foams RPUF-6 and RPUF-1 to control the nucleation of bubbles, it seems that it was not enough to yield homogenous cells. This is particularly evident for RPUF-11, which also included a higher amount of catalyst, thus, the foaming process might have been too fast, and the amount of surfactant did not suffice to control the cellular growth.

### 3.2. Optimization of Formulations

Since the prediction profiler (Figure 1) was obtained from the reduced model, two random additional formulations were chosen to validate the reduced models to confirm the model’s reliability at the 95% confidence level as depicted in Table 5. According to the prediction profiler, using the desirability function, the formulation corresponding to the predicted responses with higher desirability was: 8:2% of BA, 0.80% of CAT, and 1.5% of SURF, which corresponds to run 1 at Table 5. Notice should be made that this formulation uses 20% less catalyst than the formulation involving the conventional polyol, and that the LBP used in this work was not purified hence, it contains residual DBU catalyst which is frequently used in the PU formulation.

### 3.3. Characterization of Bio-Based Rigid Polyurethane Foams

Besides the determination of the thermal conductivity and density, the morphology, closed cell content, thermal stability, mechanical and viscoelastic properties of the bio-based RPUF produced using the optimized formulation were also evaluated and compared with those of the foam produced using conventional polyol (RPUF-conv) and of a commercial RPUF (RPUF-commercial). Additionally, the fire reaction performance of bio-based RPUF was assessed.

From the results presented in Table 6, the bio-based RPUF prepared using solely crude LBP presents the lowest density (33.2 kg/m^3^). Consequently, the mechanical properties are also reduced since the density is correlated with mechanical properties [37,46]. Comparing the bio-based RPUF with the RPUF-conv, which have different densities, the better mechanical properties of RPUF-conv can be correlated with its density (44.5 kg/m^3^) deriving from the higher content of solid phase. In this case, the higher thickness and length of cell wall/struts contribute to a better load-bearing capacity of the cell walls [47]. Yet, other aspects associated with the morphology of the cells, such as cell size distribution and cell shape [48,49], may also contribute to the lower mechanical performance of the bio-based RPUF. Nevertheless, its compressive stress value (σ10) is 127 kPa (Figure 3), i.e., above 100 kPa which is the minimum value required for RPUFs to be used as thermal insulation material in buildings [50]. However, as discussed ahead, should the application involve higher temperatures, the mechanical performance of this bio-based RPUF may be compromised. 

Regarding the thermal conductivity, the bio-based RPUF presented a slightly higher value than the RPUF-commercial, yet it is in the range of the market requirements. On the other hand, this value is lower than that of RPUF prepared using the conventional polyol. It is well known that the biggest contribution (>60%) to the total thermal conductivity value derives from the blowing agent (via gas conduction) following the traditional heat transfer mechanisms [21,51]. Since both, the bio-based RPUF and RPUF-conv were formulated with a blend of *n*-pentane/water, the difference in thermal conductivity can be attributed to the influence of the polyol on the average cell size. In fact, the average cell size of bio-based RPUF is around 28% smaller (nearly 150 µm less) than that of RPUF-conv and consequently it may influence the reduction in thermal conductivity. The earliest heat transfer models consider that conduction is the most significant mode of heat transfer when the cells have a uniform size distribution. However, recent models for thermal conductivity consider that other parameters such as the average and distribution of cell size as well as the cell wall thickness significantly impact the thermal conductivity [52]. Regarding the values of closed cell content presented in Table 6, that of the bio-based RPUF is 22% lower than the RPUF-commercial. In turn, the RPUF-conv has a similar content of closed cell (67.9%) to that of the bio-based RPUF but higher thermal conductivity, possibly due to the larger average cell size, as discussed before.

Comparing the morphology of the bio-based RPUF, RPUF-conv, and RPUF-commercial, presented in Figure 4, it can be seen that all foams present essentially a hexagonal closed cell structure. Although the average cell size of the bio-based RPUF (248 µm) is similar to that of the RPUF-commercial (221 µm), the size distribution is broader. This can be attributed to the high viscosity and reactivity of LBP despite the presence of PC-oligomer, as previously discussed. As regards the high viscosity of polyol it can reduce Ostwald ripening, thus increase the number of cells and consequently reduce the average cell size of RPUF. Yet, high values of viscosity can also cause coarser and less uniform cells [20,53]. In addition, because the crude LBP contains residual DBU catalyst this can increase the velocity of the blowing reaction, yielding bigger cells with a broader cell size distribution. This trend is well known as the cells of bio-based foams present greater heterogeneity when compared to commercial RPUFs [41,54,55]. Song et al. [56] clarified the importance of controlling the viscosity of the reactant mixture during the foaming process and according to these authors, very low viscosity can lead to rapid floating of bubbles and too high a viscosity can suppress the formation and the growth of bubbles. As regards the closed cell content, the bio-based-RPUF and RPUF-conv presented similar values (around 68%) but smaller than the RPUF-commercial (86.5%). These results are in agreement with the SEM micrographs (Figure 4), which show that the bio-based RPUF presents a more irregular cell structure than the RPUF-commercial, and a smaller value for closed cell content. This may be attributed, again, to the characteristics of the crude LBP, i.e., its viscosity, presence of residual catalyst, and of PC-oligomers. Indeed, a review of Peyrton & Avérous et al. [20] remarked on the fact that the small molar mass molecules in polyols can act as plasticizer increasing the drainage and open cell content. Since LBP has a relatively high content of aromatic moieties which can form hard domains and the PC-oligomer is aliphatic thus can form flexible domains, microphase segregation can occur. This can subsequently lead to drainage resulting in cell rupturing due to the extensional thinning and stress caused by the high concentration of hard domains thought to be associated with hydrogen bonding between the isocyanate and the aromatic moieties of lignin [57,58,59]. In addition, as mentioned above, the presence of residual catalyst in the crude LBP affects the kinetics of RPUF reactions and consequently affects the closed cell content. In fact, it is known that RPUF made using bio-based polyol present lower closed cell content compared to RPUF made using conventional polyol [60,61,62].

In summary, the density, thermal conductivity, and average cell size values of the bio-based RPUF are in the range of the RPUF-commercial, only the closed cell content was around 21% lower than the value of the RPUF-commercial. Furthermore, these values are lower than the values reported in the literature for RPUF based on LBP (also derived from kraft lignin) with similar viscosity [42,63].

In order to assess the impact of crude LBP on the viscoelastic properties of the optimized bio-based RPUF the storage modulus and tan (δ) were measured and compared with those of the RPUF-conv and RPUF-commercial. Figure 5a shows, that the bio-based RPUF presents a much lower modulus than the RPUF-commercial, and RPUF-conv has an intermediate behavior. This tendency is in agreement with what was registered in the stress-strain tests. The lower modulus of the bio-based RPUF may be associated with the composition of the crude polyol, namely the presence of the PC-oligomer which tends to act as plasticizer, as well as the lower solid content and morphology of the cells. Furthermore, as mentioned above, at higher temperature, the performance of the bio-based foam many be limited. Indeed, some studies have shown that the increase in temperature, decreased the mechanical properties of RPUF (compressive strength and modulus of elasticity), i.e., an increase of 20 °C can reduce these properties by up to 10% [64,65]. Considering that the bio-based RPUF presents relatively low mechanical properties, it can be a drawback if this foam is subjected to relatively high in-service temperatures. Yet, the optimized bio-based RPUF presents the highest T_g_ (185 °C) when compared to the RPUF-commercial (163 °C) and RPUF-conv (88 °C). Hence, despite the plasticizing effect of the PC-oligomer, the highly aromatic backbone of LBP seems to surpass its effect hindering the mobility of PU chains. However, the peak of the optimized bio-based RPUF is wider than the peaks of the other foams, indicating network heterogeneity. In fact, the width of the tan δ curve can be considered a good indicator of network homogeneity/heterogeneity, while the peak height can indicate the elasticity of the polymer suggesting that the RPUF-commercial is more elastic [39,66]. Indeed, this is in agreement with the higher value of Young’s modulus obtained for RPUF-commercial.

The thermo-oxidative stability of the bio-based RPUF was also compared with that of the RPUF-commercial and of the RPUF prepared using a conventional polyol. Figure 6 displays the thermograms of those samples as well as those of KL, LBP, and pMDI for comparison purposes. The degradation of the bio-based PUFs is characterized by three main degradation steps: (i) below 200 °C attributed to the labile moieties of LBP and PC oligomers, (ii) between 200 and 300 °C assigned to the thermal degradation of urethane moieties, and (iii) above 400 °C related to the thermal decomposition of the ether linkages of the polyol segments and char formation [67,68,69]. In turn, the degradation of RPUF-commercial only starts at 265 °C, which reveals the lower thermal stability of the bio-based RPUF, which can be attributed to the use of crude LBP. In fact, the presence of PC-oligomers and thermally labile moieties of LBP decreases the thermal stability of the PUF produced thereof [70]. Although KL presents higher thermal stability due to its natural aromatic structure, the LBP presents lower thermal stability due to the presence of PC-oligomer formed during the alkoxylation as a by-product. A review by Sen et al. [71] provides a comprehensive overview that discusses extensively the thermal properties of lignin and derivatized lignin in copolymers, blends, and composites. Accordingly, the alkylation and alkoxylation of lignin significantly increases the thermal degradation when compared to underivatized lignin. After derivatization, such as alkylation, the intramolecular hydrogen bonds of lignin are destroyed, whereas, upon alkoxylation, the acidic phenolic OHs are replaced by comparatively less acidic secondary aliphatic OHs. This reduces the potential to create strong intramolecular hydrogen bonds and consequently reduces their thermo-oxidative stability [71,72,73].

Despite the lower thermo-oxidative stability of the bio-based RPUF, in view of the considerable content of lignin it was expected that this foam would have a better fire behaviour when compared to the RPUF prepared using a conventional polyol. Indeed, RPUF can easily ignite and burn when exposed to a heat source releasing very toxic gases [74]. Hence, usually flame retardants such as halogenated paraffins and phosphorus containing compounds have to be added to the formulation [75]. In turn, lignin has been reported as a potential flame retardant due to its aromatic structure which is able to produce a large amount of char residue upon heating at elevated temperature, reducing the combustion heat [76,77]. Yet, most of the studies reporting the flame retardancy of lignin involved its direct addition to the polymer matrix together with conventional flame retardants and a synergistic effect between both, or the use of lignin chemically modified with other traditional flame retardants such as organophosphorus and phosphorus-nitrogen compounds [78,79,80,81]. 

To evaluate the fire reaction performance of the bio-based RUF prepared using crude LBP and to compare it with the RPUF prepared using conventional polyol without flame retardant, preliminary burning tests were carried out, which showed that the bio-based RPUF takes 3 s more to burn (25%) than RPUF-conv which amounts to a 25% increase in time to burn (see Appendix A). Furthermore, the heat release rate (HRR) being one of the most important variables regarding the flammability of materials, which corresponds the quantity of heat generated per unit area and time, expressing the intensity of fire, the samples were analyzed by cone calorimetry [75]. In turn, the maximum average rate of heat emission (MARHE) is a single number used to describe the maximum heat released between the start and the end of test, defined as the cumulative heat emission in the test period divided by the time, e.g., some materials must meet the EN45545 standard (European railway standard for fire safety), which requires a MARHE value lower than 90 kW/m^2^ [82]. Figure 7 depicts the samples before and after the cone calorimetry test. The results presented in Table 7 show that the values of HRR and MARHE determined for the bio-based RPUF are lower than those obtained for the RPUF prepared using the conventional polyol, thus confirming the flame retardancy effect of lignin, and meeting the EN45545 standard requirement. Our results are in agreement with the previous study by Pinto et al. [42] who observed the decrease in MARHE in RPUF using LBP obtained from oxyalkylation of lignin with PO. In contrast, the results of the study by Duval et al. [15] showed that the crude LBP obtained from oxyalkylation using ethylene carbonate did not have any flame retardant effect in RPUF. The flame retardancy effect of lignin-based polyol in the PUFs is not clearly defined yet, since in the literature there are only a few studies regarding the fire properties of these types of materials. Nevertheless, it should be stressed that in the Duval’s work only 25% of the polyol was derived from lignin. Moreover, the oxyalkylation of lignin was carried out using ethylene carbonate. Accordingly, the double bonds of the ethylene moieties of the carbonate may have also played a role in the burning process, limiting the positive impact of lignin. 

## 4. Conclusions

This work demonstrated the potential of the crude LBP, containing 20 wt % of lignin, as a sustainable alternative to conventional polyol in the production of bio-based RPUF for use as a lightweight thermal insulation material. The application of a DoE provided important information about the effect of BA (mixture of n-pentane and water), surfactant, and catalyst content on thermal conductivity and density of RPUFs. Among the examined properties, only density proved to be highly influenced by the BA. Optimization of the formulation afforded a bio-based RPUF with low thermal conductivity (0.029 W/m·K) and density (33.2 kg/m^3^) which meets the market requirements for thermal insulation panel. Additionally, this optimization process allowed a reduction in the amount of catalyst used in 20%. Moreover, the bio-based RPUF showed a value of the average cell size close to RPUF-commercial. On other hand, due to the characteristics of crude LBP such as high reactivity and the presence of PC-oligomers the morphology of the ensuing bio-based RPUF is heterogeneous concerning cell size, and has a lower number of closed cells. These aspects compromised, to a certain extent, the mechanical performance of the bio-based foam and its use at higher temperatures may be limited. Nevertheless, the compressive stress resistance of the bio-based RPUF of 127 kPa is higher than the value required for thermal insulation materials in civil construction (100 kPa). Although the bio-based RPUF presented lower thermo-oxidative resistance compared to the RPUF-conv, its fire reaction performance was better and complies with the EN45545 standard requirement. Overall, the potential of crude LBP to replace the conventional polyol was demonstrated, proving that the optimization of formulations is an important step in achieving the technical requirements of the final product. However, the shortcomings need to be addressed in future research to further improve the potential of LBP in the production of RPUF. The novelty of this study is the use of 100% crude LBP obtained via the oxyalkylation of LignoBoost kraft lignin in the production of RPUF which presented improved fire resistance.

## Figures and Tables

**Figure 1 polymers-15-01074-f001:**
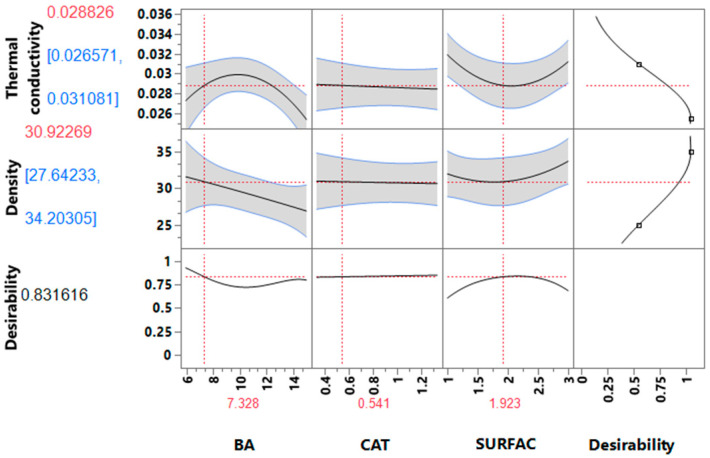
Prediction profiler for thermal conductivity and density of RPUFs.

**Figure 2 polymers-15-01074-f002:**
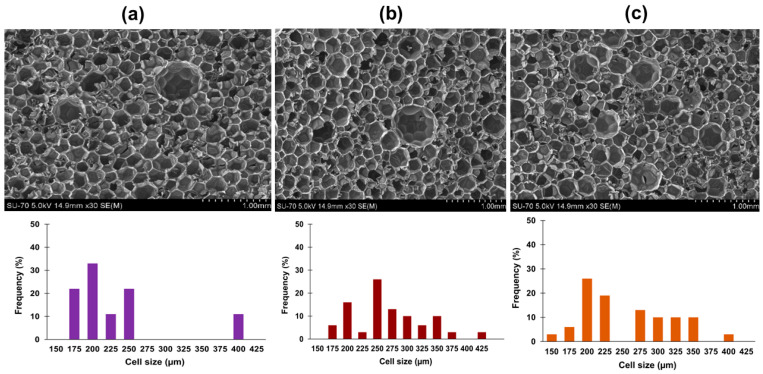
SEM micrographs and cell size distribution in RPUF-1 (**a**), RPUF-6 (**b**), and RPUF-11 (**c**). Free ascent direction and a 30× magnification were used.

**Figure 3 polymers-15-01074-f003:**
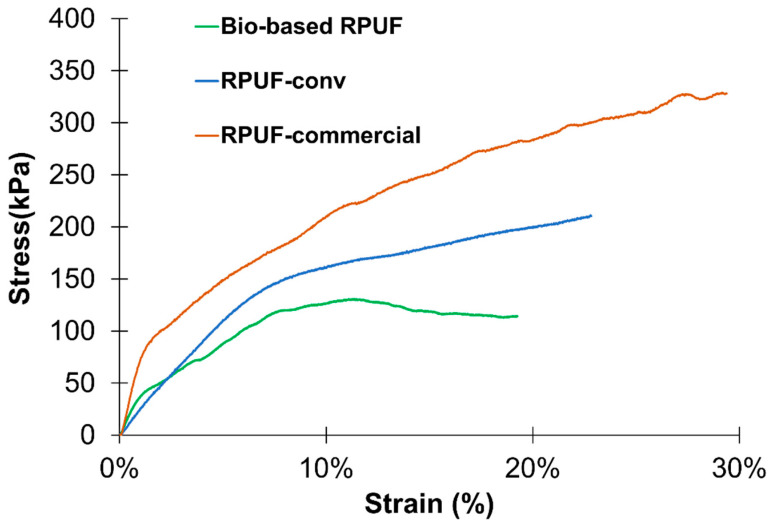
Compressive stress-strain curves of bio-based RPUF, RPUF-conv, and RPUF-commercial.

**Figure 4 polymers-15-01074-f004:**
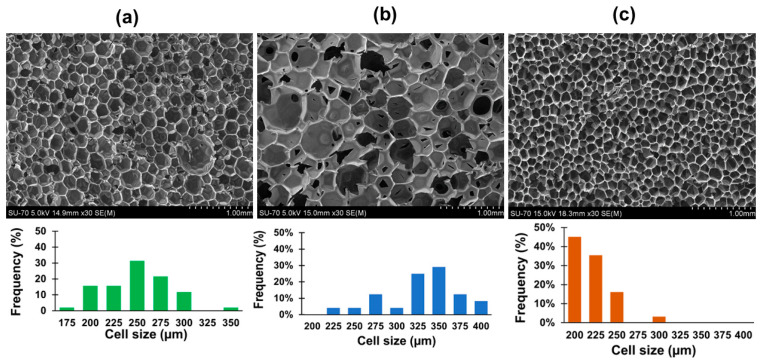
SEM micrographs and cell size distribution of bio-based RPUF (**a**), RPUF-conv (**b**) and RPUF-commercial (**c**). Free ascent direction and a 30× magnification were used.

**Figure 5 polymers-15-01074-f005:**
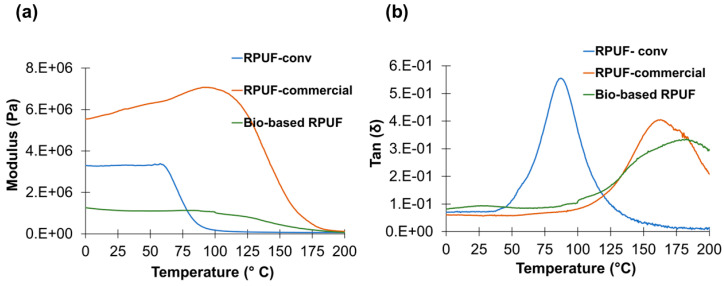
Thermo-mechanical properties of optimized bio-based RPUF, RPUF-conv, and RPUF-commercial (**a**) storage modulus curve, (**b**) tan δ curve at 1 Hz.

**Figure 6 polymers-15-01074-f006:**
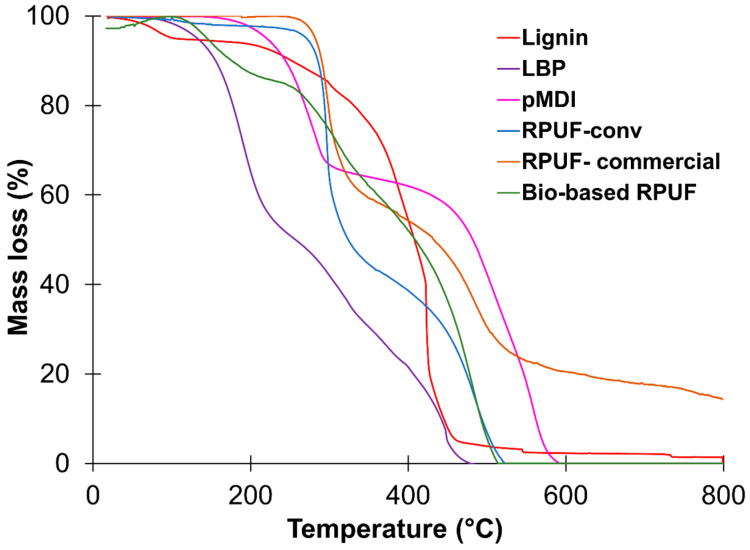
Mass loss of RPUF, lignin, LBP, and pMDI as a function of temperature.

**Figure 7 polymers-15-01074-f007:**
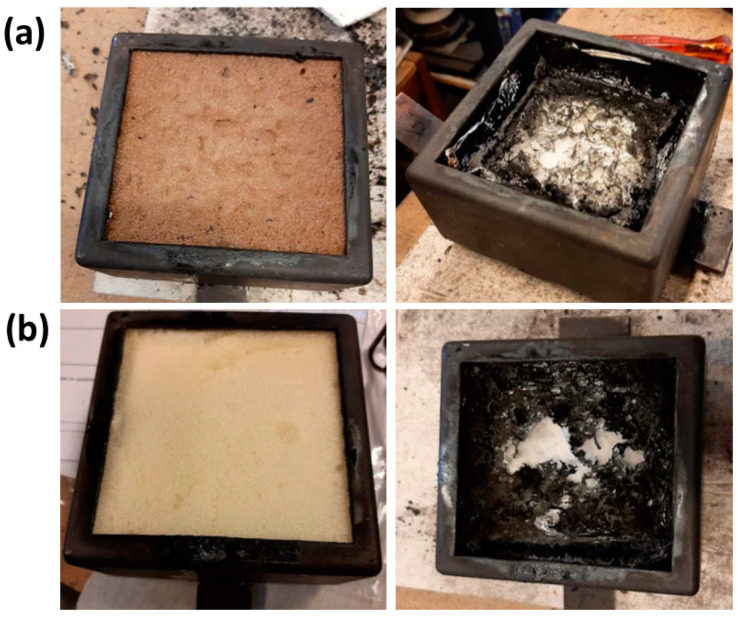
(**a**) bio-based RPUF and (**b**) RPUF-conv before (**left**) and after (**right**) burn.

**Table 1 polymers-15-01074-t001:** Formulations of the RPUFs based on parts per hundred of polyol by weight (php).

Foams Codes	Polyol	pMDI	Surfactant	Catalyst	Blowing Agent	
					*n*-Pentane	Water
RPUF-1	100	93.7	1	0.5	7.0	2.0
RPUF-2	100	93.7	3	0.5	7.0	2.0
RPUF-3	100	93.7	1	1.5	7.0	2.0
RPUF-4	100	93.7	3	1.5	7.0	2.0
RPUF-5	100	93.7	1	0.5	15	2.0
RPUF-6	100	93.7	3	0.5	15	2.0
RPUF-7	100	93.7	1	1.5	15	2.0
RPUF-8	100	93.7	3	1.5	15	2.0
RPUF-9	100	93.7	2	1.0	7	2.0
RPUF- 10	100	93.7	2	1.0	15	2.0
RPUF-11	100	93.7	2	0.5	11	2.0
RPUF-12	100	93.7	2	1.5	11	2.0
RPUF-13	100	93.7	1	1.0	11	2.0
RPUF-14	100	93.7	3	1.0	11	2.0
RPUF-15	100	93.7	2	1.0	11	2.0
RPUF-16	100	93.7	2	1.0	11	2.0
RPUF-conv	100	93.7	1.5	1.0	8	2.0

**Table 2 polymers-15-01074-t002:** Experimental data for the 16 foams formulations.

Foams Codes	Thermal Conductivity at 25 °C, W/m·K (Mean ± * SD)	Density, kg/m^3^ (Mean ± * SD)	Visual Inspection
RPUF-1	0.0313 ± 9.00 × 10^−5^	31.91 ± 1.85	rigid, good dimensional stability
RPUF-2	0.0312 ± 1.10 × 10^−4^	36.20 ± 1.90	slight shrinkage, rigid
RPUF-3	0.0314 ± 8.00 × 10^−4^	34.45 ± 2.40	slight shrinkage, rigid
RPUF-4	0.0298 ± 2.00 × 10^−4^	32.87 ± 2.95	rigid, good dimensional stability
RPUF-5	0.0288 ± 7.00 × 10^−5^	29.91 ± 2.90	slight shrinkage
RPUF-6	0.0279 ± 1.40 × 10^−4^	27.61 ± 1.00	rigid, slight shrinkage
RPUF-7	0.0308 ± 3.40 × 10^−5^	26.10 ± 1.00	some holes
RPUF-8	0.0329 ± 3.40 × 10^−4^	23.62 ± 2.00	less rigid, some holes
RPUF-9	0.0297 ± 9.00 × 10^−5^	27.55 ± 2.80	less rigid, good dimensional stability
RPUF- 10	0.0264 ± 1.10 × 10^−4^	26.12 ± 1.00	rigid, some holes
RPUF-11	0.0281 ± 8.00 × 10^−5^	28.21 ± 1.00	rigid, good dimensional stability
RPUF-12	0.0313 ± 2.80 × 10^−5^	27.33 ± 2.00	rigid, slight shrinkage
RPUF-13	0.0348 ± 1.30 × 10^−4^	28.60 ± 2.60	rigid
RPUF-14	0.0314 ± 2.40 × 10^−4^	28.93 ± 2.90	less rigid
RPUF-15	0.0316 ± 2.10 × 10^−4^	30.40 ± 2.10	rigid, slight shrinkage
RPUF-16	0.0306 ± 2.10 × 10^−4^	29.40 ± 1.95	rigid, some holes

* Standard deviation (SD).

**Table 3 polymers-15-01074-t003:** ANOVA results for the reduced regression models for thermal conductivity and density.

Source	Responses					
	Thermal Conductivity			Density		
	DF	SS	MS	DF	SS	MS
Model	5	0.00004928	9.856 × 10^−6^	5	115.040	23.008
Error	10	1.8657 × 10^−6^	1.8657 × 10^−6^	10	44.129	4.412
Total	15	0.00006794	-	15	159.170	-
*F ratio*	5.283			5.213		
*p*-value	<0.0124			<0.0130		
R^2^	0.730			0.722		
R^2^ adjusted	0.588			0.584		
Mean of response	0.0304			29.30		

DF: degree of freedom; SS: sum of square; MS: mean of square.

**Table 4 polymers-15-01074-t004:** Characteristics of the selected foams based on visual inspection.

Properties	RPUF-1	RPUF-6	RPUF-11
* Formulation	1.0 Surf/0.5 cat/7:2 BA	3.0 Surf/0.5 cat/15:2 BA	2.0 Surf/1.5 cat/15:2 BA
Density, kg/m^3^	31.9 ± 2.50	27.6 ± 1.30	28.2 ± 1.70
Thermal conductivity, W/m·K	0.0313 ± 2.00 × 10^−4^	0.0279 ± 1.2 × 10^−4^	0.0281 ± 3.8 × 10^−4^
Cell size average, µm	232 ± 82	266 ± 57	254 ± 69
BET surface area, m^2^/g	5.96 ± 1.20	4.30 ± 0.90	4.81 ± 0.75
Pore volume, cm^3^/g	9.70 × 10^−3^	9.62 × 10^−3^	4.90 × 10^−3^

* Formulation codes: surf—surfactant; cat—catalyst, BA—blowing agent.

**Table 5 polymers-15-01074-t005:** Validation of the fitted model for I_OH_ and viscosity.

Run	Variables				Responses		
	BA, %	CAT, %	SURF, %	Predicted TC, W/m·K	Experimental TC, (Mean ± SD)	Predicted Density, kg/m^3^	Experimental Density, kg/m^3^ (Mean ± SD)
1	8:2	0.80	1.5	0.0300 (0.0286–0.0312) *	0.0290 ± 3.18 × 10^−4^	31.93 (28.9–34.7) *	33.1 ± 1.00
2	5:2	1.0	1.5	0.0250 (0.0221–0.0 293) *	0.0292 ± 4.38 × 10^−4^	32.90 (27.3–38.5) *	28.5 ± 1.10

BA—blowing agent, CAT—catalyst, SURF—surfactant; TC—thermal conductivity; SD—standard deviation. * Values within of 95% confidence interval.

**Table 6 polymers-15-01074-t006:** Average values of the characteristics of ensuing RPUF.

Properties	Bio-Based RPUF	RPUF-Conv	RPUF-Commercial
Density, kg/m^3^	33.2 ± 1.00	44.5 ± 2.80	34.0 ± 1.00
Thermal conductivity, W/m·K	0.029 ± 3.18 × 10^−4^	0.034 ± 1.90 × 10^−4^	0.024 ± 1.65 × 10^−4^
Cell size average, µm	248 ± 88.0	344 ± 108	221 ± 69.0
Closed cell content, %	68.1 ± 1.50	67.9 ± 1.62	86.5 ± 0.90
Compressive stress σ 10%, kPa	127 ± 9.50	160 ± 13.0	209 ± 18.0
Young’s modulus, kPa	1954 ± 180	2449 ± 210	7645 ± 253

**Table 7 polymers-15-01074-t007:** Reaction to fire parameters of bio-based RPUF and RPUF-conv.

Properties	Bio-Based RPUF	RPUF-Conv
Average HRR_60seconds_, kW/m^2^	20.9 ± 2.10	72.8 ± 6.20
Average HRR_300seconds_, kW/m^2^	23.9 ± 4.55	29.3 ± 2.20
MARHE, kW/m^2^	33.1 ± 3.60	110.8 ± 9.50

## Data Availability

Not applicable.

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
