# Peer review of "Bio-Based Polyurethane Foams from Kraft Lignin with Improved Fire Resistance"

_polymers, 2023, doi:10.3390/polym15051074_

Round 1

Reviewer 1 Report

This paper illustrates the possibility of using crude biobased ligning derivatives as substitutes for conventional polyols in the synthesis of rigid polyurethane foams.
The results are interesting and the article is quite well written.

There are however few points that must be improved:

Line 126 was Alcupol used for  the conventional polyether polyol foam?

Line 139 It is nor clear when polyol was added in component A

Table 1 how the composition of RPUF-conv was selected?

Line 440 et following. I do not agree with the interpretation of the TGA results. Polyether bond is quite thermally unstable and its breakdown should occur at lower T than that of urethane bonds The step starting at 128°C can be attributed to ther low molecular weight PC-Oligomers evaporation

Line 450 Decreases instead of increases

Line 460 thermo-oxidative instead of thermos-oxidative

Figure S4 why some values are in red? Please explain in the caption the *appearing  in the table

Author Response

Dear reviewer,

Reviewer 2 Report

This study is a "good one" from the point of view of originality and scientific level which is deserving attention. This study will be a good one for other groups working on the similar matter of subjects, to be able to carry out similar and possibly further studies in the future. Before publication, the following suggestions are strongly recommended:

1) A brief review about the use of polymeric foams in engineering applications it is strongly suggested.

2) The authors should extend the literature review about the mechanical and thermal behaviour of polyurethane foams when exposed to high temperatures. Please mention in % the variation with temperature of the thermo-mechanical and thermos-physical properties of PUR foam materials. Check this: https://doi.org/10.1177/10996362211050919

DOI: 10.1186/s40038-016-0012-3

3) At the end of the introduction section, the aim of study should be more emphasized.

4) How the authors defined the loading rate in the mechanical tests? In my opinion 10 mm/min is a bit too high.

5) How the heating rate in the TGA tests was defined? This parameter strongly influences the decomposition temperature.

6) Can the authors provide more information about the methodology used to perform the DMA tests?

7) “From the results presented in Table 6, the bio-based RPUF prepared using solely 333 crude LBP presents the lowest density. Consequently, the mechanical properties are also 334 reduced since the density is correlated with mechanical properties [34,43].”  The authors should provide more information about this, mentioning the fact that the higher strength in high-density foams can be associated with the higher % of solid contributing to the load-bearing capacity of the cell wall.

8) “. Yet, the optimized bio-based RPUF presents the highest Tg (185 °C) when compared to the RPUF-commercial (163 °C) and RPUF-conv (88 °C).” How these Tg values were defined? The authors should check that the Tg was defined using the same criteria.

9) Overall, The discussion of the results must be improved; authors should emphasize the implications of their findings.

10) In the conclusions, the authors should explain the significance and shortcomings of the research work, instead of repeating the results obtained before.

Author Response

Dear reviewer,

Reviewer 3 Report

1.    What are the statistics related to different characterization results  such as cell size distribution, mechanical testing, BET, cone calorimeter ? How many times was each characterization repeated? What was the sample size?

2.       Provide the statistics in Table 4,6 and 7 for different characterization results.

3.       How was the formulation of RPUF-conv selected? Is there any reference for choosing the particular composition of the ingredients in RPUF-conv?

4.       Provide reference for the foam preparation method.

5.       What are the time scales of the burning experiments as shown in the  supplementary video. Provide time value for the two samples.

6.   Page 1, Lines 20-21: The abstract should include the statistics related to the results described.

7.  Page 1, Lines 24-25: Please provide the quantitative numbers as described in the main text here to depict the better fire performance of bio-based PUR.

8.  Provide source of n-pentane.

9.  The  scale bars of SEM images are not visible.

Author Response

Dear reviewer,

Round 2

Reviewer 2 Report

The paper is significantly improved following the review process. The authors responded to all the reviewers' comments, therefore, I recommend that the paper be published in the current form. Of course, the final decision belongs to the Editor.

Reviewer 3 Report

The authors have addressed to the reviewer's comments.